# A Functional Magnetic Resonance Imaging Investigation of Hot and Cool Executive Functions in Reward and Competition

**DOI:** 10.3390/s25030806

**Published:** 2025-01-29

**Authors:** Hsin-Yu Lin, Hoki Fung, Yifan Wang, Roger Chun-Man Ho, Shen-Hsing Annabel Chen

**Affiliations:** 1Centre for Research and Development in Learning, Nanyang Technological University, Singapore 637335, Singapore; linhy@ntu.edu.sg; 2Department of Psychology, School of Social Sciences, Nanyang Technological University, Singapore 639818, Singapore; hokifung@berkeley.edu; 3Department of Psychiatry and Biobehavioral Sciences, Semel Institute for Neuroscience and Human Behavior, University of California, Los Angeles, CA 90095, USA; 4Key Laboratory of Modern Teaching Technology, Ministry of Education, Shaanxi Normal University, Xi’an 710062, China; wangyifan0929@126.com; 5Department of Psychological Medicine, Yong Loo Lin School of Medicine, National University of Singapore, Singapore 119228, Singapore; pcmrhcm@nus.edu.sg; 6Lee Kong Chian School of Medicine, Nanyang Technological University, Singapore 636921, Singapore

**Keywords:** hot and cool executive functions, fMRI, social competition, reward sensitivity

## Abstract

Social and environmental influences are important for learning. However, the influence of reward and competition during social learning is less understood. The literature suggests that the ventromedial prefrontal cortex is implicated in hot executive functioning (EF), while the dorsolateral prefrontal cortex is related to cool EF. In addition, reward processing deficits are associated with atypical connectivity between the nucleus accumbens and the dorsofrontal regions. Here, we used functional magnetic resonance imaging (fMRI) to determine the role of hot and cool EF in reward processing and their relationship to performance under social competition. We adapted a reward-based n-back task to examine the neural correlates of hot and cool EF and the reward influence on performance during competition. A total of 29 healthy adults showed cortical activation associated with individual differences in EF abilities during fMRI scans. Hot and cool EF activated distinct networks in the right insula, hippocampus, left caudate nucleus, and superior parietal gyrus during the no-competition task, while they differentially activated the right precuneus and caudate nucleus in the competition condition. Further analysis revealed correlations between the Hot–Cool network and reward sensitivity and risk-taking behaviour. The findings provided further insights into the neural basis of hot and cool EF engagement in the socio-emotional regulation for learning.

## 1. Introduction

Increasing evidence highlights the importance of social and environmental influence on individual differences in learning [1,2,3]. Learning is a process by which a social group transmits knowledge and skills to its members, involving social competition and social rewards brought about by the social group [4,5,6,7]. Under social interaction, executive functions perform dynamic changes and development influenced depending on environmental changes and external stimuli. Executive functions, such as affective evaluation and cognitive control, could be modulated by socio-emotional regulation related to reward under inter-group competition [8,9,10,11]. Exploring the neural basis of the synergistic engagement of executive functions is significant for teaching strategies and learning interventions from the perspective of social interaction.

Emerging evidence in executive functions or cognitive control suggests that this top-down neural cognitive construct can be divided into two distinct but interacting components: cool executive function (cool EF) and hot executive function (hot EF). Cool EF is thought to be engaged by abstract problems, such as number processing, sorting, and rule use, whereas hot EF is related to stimuli and outcomes that are emotionally salient [12,13]. Hot and cool executive networks help individuals regulate and balance the processing of social rewards, leading to different internalizing and externalizing behaviours [14,15,16]. Deficits in both hot and cool EF are strongly associated with abnormal behavioural response patterns [17,18,19]. Previous studies engaging the executive functions, such as response inhibition [20,21,22,23], cognitive flexibility [10,24], attention [25], or working memory [26,27], have shown performance to differ under different reward conditions (i.e., with vs. without reward incentives). However, these studies have mostly examined cool and hot EF in isolated conditions. This limits our understanding of how the cortical activity related to hot and cool EF interacts to modulate behavioural responses during the “hot” context of emotional involvement and reward feedback.

Hot EF modulates negative emotional arousal and enhances motivation in emotionally salient contexts, while cool EF regulates goal-oriented and flexible switching behaviours [15,19,28]. The two aspects of executive functioning are thought to be associated with distinct regions of the prefrontal cortex [19,29,30,31,32]. Cool EF is generally linked to the dorsolateral prefrontal cortex (DLPFC), whereas the ventromedial prefrontal cortex (VMPFC) is thought to govern the top-down process in hot EF [32,33,34,35]. The n-back task has consistently been reported to activate brain regions associated with cool EF network, such as the dorsal anterior cingulate cortex, DLPFC, superior parietal lobule (SPL, BA 7) and the cerebellum [27,36]. As for reward processing, sensitivity to hot EF has been linked to activation of the mesocorticolimbic reward circuitry, including the amygdala, nucleus accumbens (NAcc), anterior cingulate cortex, and orbitofrontal cortex during viewing or anticipation of reward stimuli [37,38,39,40].

Cognitive control has also been studied in relation to reward processing. The key cortico-basal ganglia reward circuitry mediates cortical signals to affect the cognitive control network that regulates different cognitive tasks [10,41,42,43]. Studies have shown that individuals with reward processing deficits exhibit atypical resting-state functional connectivity between the NAcc and dorsofrontal regions involved in cognitive control [44]. Additionally, individual differences in resting-state functional connectivity between the NAcc and the DLPFC have been found to correlate with individual differences in behavioural measures of cool executive control [44,45]. Furthermore, structural imaging data have shown that increased striatal connection strength with the DLPFC is associated with being patient, whereas increased striatal connection strength with the VMPFC is associated with more impulsive behaviour during a delay-discounting reward task [46,47]. This finding may suggest that both striatal tracts are associated with executive functioning abilities in reward processing.

Reward usually operates in social competition contexts. Previous studies have shown that brain regions associated with cognitive networks, such as the medial prefrontal cortex (mPFC), exhibit increased activation in social competition, potentially depriving the attentional resources associated with the completion of working memory, and thus impairing task performance [48,49]. Meanwhile, better working memory performance under social competition is associated with greater bilateral striatal inhibition [50]. In addition, recent studies on individuals with autism spectrum disorder have found that those with social reward processing deficits may show dysfunction in this circuitry [51,52,53,54]. In reward processing, risk-taking is also considered an important element. Previous studies have demonstrated how decision-making is influenced by the interplay between potential rewards and the perceived level of risk involved [37,55]. Furthermore, research has highlighted the cognitive processes underlying risk preferences and their implications for behaviour, and their association with neural responses in the brain [25,56,57,58].

Although there are fMRI studies examining hot and cool EF [27,59], no study has yet determined how the hot and cool networks are associated with the influence of reward on executive control and its relationship with cognitive performance, particularly in the context of social competition. The present study integrated real monetary rewards within a task-based fMRI design to investigate these associations, providing a more direct and tangible incentive for the participants [26,38,60,61]. We examined functional activation patterns of social reward processing in healthy adults using task-based fMRI, a non-invasive neuroimaging technique that measures brain activity by detecting changes in blood oxygenation levels, known as the blood oxygenation level dependent (BOLD) signal. This technique allows for the detection of neural activity in response to cognitive tasks, facilitating the investigation of executive functions and reward processing. We also explored the neural correlates of behavioural measures of hot and cool EF and reward sensitivity. Specifically, we incorporated multiple behavioural measures including the Behaviour Rating Inventory of Executive Function—adult version (BRIEF-A) to measure hot and cool EF, Go/No-Go task for cool EF, and sensitivity to reward, as well as risk-taking, as measured by the Balloon Analogue Risk Task (BART) for Hot–Cool EF. We hypothesized that the following: (1) Healthy adults will show different activations between hot and cool EF and the neural mechanism may be further affected by the competition factor; (2) Hot EF areas will be correlated with Emotional Control and Self-Monitor domains of BRIEF-A; (3) Cool EF areas will be associated with Working Memory, Inhibit, and Shift domains of BRIEF-A [62] and Go/No-Go task; (4) Hot–Cool EF areas will be correlated with risk-taking (BART) and Sensitivity to Reward to further understand the reward effect.

## 2. Materials and Methods

### 2.1. Participants

A total of 33 right-handed healthy adults, aged between 21 and 40, were initially recruited through poster advertisements at Nanyang Technological University and National University of Singapore. This study was approved by the respective institutional review boards (NHG DSRB Ref 2017/01125; IRB-2016-01-003) in these two universities. All participants provided written informed consent before participation and were screened to rule out (1) neurological or psychiatric disorders, (2) irremovable metallic objects or implants (e.g., pacemaker), and (3) not taking any medication at the time of the study or other factors that affect the result or increase the risk of having an adverse event during an MRI scan. Four subjects were excluded from the analyses due to scan incompletion, incidental findings (e.g., arachnoid cysts) or excessive motion artifacts, and 29 healthy participants (Male/Female: 14/15, Age Mean/Std = 25.55/4.54) were included in the final analyses. The participants received a reimbursement fee for their time participating in the study, along with an additional monetary reward based on their task performance. The demographic information for the subjects is listed in Table 1.

### 2.2. Behavioural Tasks

The participants completed the following behavioural questionnaires and cognitive tasks outside the scanner.

#### 2.2.1. Behaviour Rating Inventory of Executive Function—Adult Version (BRIEF-A)

The BRIEF-A is a standardized rating scale developed to provide insight into everyday behaviours associated with specific domains of the executive functions in adults aged 18 to 90 years [63]. It consists of a self-report form and an informant report form, each having 75 items in nine non-overlapping scales (Inhibit, Shift, Emotional Control, Self-Monitor, Working Memory, Initiate, Plan/Organize, Task Monitor, and Organization of Materials) as well as two summary index scales: Behavioural Regulation Index (consists of Inhibit, Shift, Emotional Control, and Self-Monitor scales), Metacognition Index (consists of Initiate, Working Memory, Plan/Organize, Task Monitor, Organization of Materials scales), and an overall functioning scale of Global Executive Composite [62]. All participants were administered the self-report form. In the current study, we focused on the Emotional Control and Self-Monitor scales, which are related to emotion regulation under the hot EF [64], and the Shift, Inhibit, and Working Memory scales, under the cool EF [65].

#### 2.2.2. The Sensitivity to Punishment and Sensitivity to Reward Questionnaire (SPSRQ)

The SPSRQ was used to quantify the participants’ sensitivity to rewards and punishments. It consists of 48 yes-or-no questions, with half of the questions assessing the participants’ sensitivity to reward and the other half assessing sensitivity to punishment. Each “Yes” response earned one point, with higher scores indicating greater sensitivity to reward or punishment. In this study, our analysis focused exclusively on the reward score of the SPSRQ, aligning with the reward model integrated into our fMRI task design.

#### 2.2.3. Go/No-Go Task

This computerized task was used to examine the participants’ cool executive functioning, particularly their inhibition ability. The participants were instructed to press the spacebar whenever they saw the letter ‘X’ and not to press any key when they saw other letters. The task consisted of 200 trials, with 20 non-‘X’ letters.

#### 2.2.4. Balloon Analogue Risk Task (BART)

The BART was used to examine the participants’ Hot–Cool EF, in particular their risk-taking tendencies and impulsivity. In this task, the participants were presented with a balloon and offered the chance to earn points by pumping 30 balloons. Each pump caused the balloon to inflate incrementally, with 50 points added to a counter for each pump, until a certain threshold was reached, at which point the balloon would be over-inflated and explode. Thus, each pump conferred a greater risk, but also greater potential reward. If the participant chose to cash out prior to the explosion, they could collect the points earned for that trial. However, if the balloon exploded, the points for that trial would be lost. The participants were not informed about the balloons’ breakpoints and the absence of this information allowed for testing both the participants’ initial responses to the task and changes in responding as they gain experience with the task contingencies.

### 2.3. fMRI Task Design

The participants took part in a competition- and reward-based letter-variant 2-back game. The game was designed to examine how (1) the magnitude of monetary reward and (2) the participation in a social competition, might affect the participants’ performance on a working memory (Cool EF) task and their corresponding brain activation. In the game, the participants were shown a series of letters, one at a time, on the screen. They were asked to indicate, starting from the third letter, whether the letter on screen was the same as the one presented two letters back. Both uppercase and lowercase letters were shown to engage the participants’ attention. However, the correct response was case-insensitive, where the letter “p” was considered the same as the letter “P” (Figure 1c). The participants performed the 2-back task under two conditions: playing alone or with a competitor. Each condition consisted of three types of trials (no-, low-, and high-reward), administered in a blocked design (Figure 1b). The participants completed two runs of each condition (approximately 6.5 min per run) for a total of four runs.

Each run started with a 6-s dummy scan and ended with a 4-s feedback screen showing their winnings for the run. There were 9 blocks of 3 reward trial types (no-, low-, high- reward) that alternated with 6-s rest blocks where a fixation cross was shown per run. There were 3 blocks of each trial type within a run (Figure 1a). The amount of money associated with the reward types were SGD0.00, SGD0.20, and SGD1.00, with the participants seeing a blank picture for no reward, a 20-cent coin for low reward, or five 20-cent coins for high reward, respectively. The participants had the opportunity to win and receive a maximum amount of SGD 14.40 as real monetary compensation. Each block lasted for 36 s, consisting of a 4-s screen at the beginning indicating the trial type and reminding the participants whether they were playing alone or with a competitor, followed by a 30-s 2-back task, and ending with a 2-s feedback screen displaying the amount of money earned for the block. The participants always completed 2 runs of each condition consecutively. The sequence of these conditions was counterbalanced across the participants.

The participant was told a cover story in which they would sometimes be playing with a competitor and at other times playing alone. During the competitor condition, they were shown a picture of their ‘competitor’ (photos were from gender-matched study team members) and told that they were competing against that previous participant and needed to achieve higher accuracy than him or her to win. In the alone condition, the participant was shown a blank grey head-profile picture and told that they needed to achieve 70% accuracy to win the block. However, in reality, there was no competitor, and the participants’ performance was evaluated the same way in both conditions, with the requirement to achieve higher than 70% accuracy to win.

### 2.4. Experimental Procedure

All participants received a 2.5-h experimental session, which comprised a 45-min behavioural testing session, a 1.5-h MRI scanning session, and a 15-min debriefing session. All participants were told a cover story in which they would be competing with another participant in the fMRI task, when in fact, there was no competitor. They had a practice session for the fMRI tasks prior to entering the scanner. During the scanning session, multimodal MRI data, including anatomical and functional MRI data, were collected. During the debriefing session, the participants were told that the competitor was actually a cover story and were administered a questionnaire to determine how much they believed in the existence of a real competitor and what strategies they used in the tasks.

### 2.5. MRI Data Acquisition

All MRI scans were performed on a 3T Siemens Prisma MR scanner with a 32-channel head coil at the Centre for Translational MR Research, National University of Singapore (NUS-TMR). High-resolution T1w images were acquired with a magnetization-prepared 2 rapid acquisition gradient echo (MP2RAGE) sequence (repetition/echo/inversion time = 5000/2.98/700 ms, field of view = 256 mm, flip angle = 4°, matrix size = 256 × 256, 176 sagittal slices, isotropic voxel size = 1 mm^3^, and no gap). Functional images were acquired in four sessions using multi-band T2*-weighted echoplanar imaging with blood oxygen level-dependent (BOLD) contrast. Each session comprised 386 volumes (repetition time = 1000 ms, slices = 64, voxel size = 3 × 3 × 3 mm^3^). Each participant’s head was immobilized with cushions inside the coil to minimize the generation of motion artifacts during image acquisition.

### 2.6. Data Analysis

In-scanner behavioural data of the 2-back task accuracy were analysed using a 2 × 3 ANOVA in IBM SPSS Statistics (Version 29.0). MRI data preprocessing and whole-brain univariate analysis were conducted using SPM 12 (Statistical Parametric Mapping software, SPM; Wellcome Department of Imaging Neuroscience, London, U.K. www.fil.ion.ucl.ac.uk/spm, version r7771) MATLAB R2020a (The MathWorks, Inc., Natick, MA, USA). The quality of images was carefully checked by the researchers to ensure there were no excessive motion artifacts (translational movement < 3 mm in any direction and rotational movement < 2°). All functional images were corrected for slice timing with the middle slice in the acquisition order as the reference, realigned to the first image to correct for head movements. The individual’s T1-weighted image was coregistered to the mean functional image and then segmented. Functional and anatomical images were normalized to the MNI (Montreal Neurological Institute) space using the DARTEL procedure [66]. A group template was first generated using the participants’ grey matter and white matter masks for more accurate inter-subject alignment. All the functional and anatomical images were registered to the group template and normalized to the MNI space. Functional images were spatially smoothed with a Gaussian kernel of 8 × 8 × 8 mm.

The preprocessed functional images were fed into the first level (subject level) setup in SPM. Each volume was mapped to one of the following possible events, task (6 conditions), and rest. A total of 12 contrasts involving different combinations of events were set up to examine the results (Appendix A). The beta from these contrasts were then brought forward to a second level (group level) analysis where we examined the contrast results as a group. Whole-brain univariate analysis with a cluster defining threshold of *p* < 0.001 (uncorrected), cluster size > 100, and family-wise error (FWE) cluster-level correction (cluster size threshold *p* = 0.01) was conducted. The locations of the significant clusters were defined by the AAL3 atlas [67]. Percent signal change of the significant clusters was then extracted using the MarsBaR toolbox [68] in SPM. and Pearson’s correlations were performed between the percent signal change and the behavioural measures. We tested the correlations between the following: (a) significant clusters in hot EF areas and the BRIEF-A Emotional Control and Self-Monitor scales; (b) significant clusters in cool EF areas and the BRIEF-A Inhibit, Shift, Working Memory scales, and Go/No-Go task; (c) significant clusters in Hot–Cool EF areas and BART and sensitivity to reward scores.

## 3. Results

### 3.1. Behavioural Measures

Table 1 shows the demographic information and behavioural measures for the 29 subjects. BRIEF-A raw scores were converted to T scores [63] according to the conversion table in the BRIEF-A professional manual. All participants’ scores were within normal range except for two participants’ T score in Emotional Control, which were considered clinically significant (90% CI = 67–77, percentile = 98%). We excluded these two participants when testing the correlation between Hot areas and BRIEF-A Emotional control. There was a ceiling effect observed in the Go/No-Go task. Consequently, we excluded this measure from the subsequent analysis.

### 3.2. fMRI Task

A two (No-Competition, Competition) × three (No-, Low-, High-Reward) ANOVA on the accuracy data of the fMRI task revealed that the effect of reward on accuracy was significant [*F*(2, 56) = 7.5, *p* < 0.01]. We applied the Bonferroni correction for multiple post hoc comparisons. There were significant differences between the No-Reward and High-Reward conditions (*p* = 0.013), and between the Low-Reward and High-Reward conditions (*p* = 0.025) (Figure 2). However, there was no statistical difference between the Competition and No-Competition conditions [*F*(1, 28) = 3.023, *p* = 0.997]. There were also no significant differences between the No-Reward and Low-Reward conditions. In order to increase the contrast, we thereby discarded Low-Reward blocks and only analysed the fMRI data between No-Reward and High-Reward in Competition and No-Competition situations.

### 3.3. Debriefing

The results of the debriefing indicated that approximately half of the participants (48.28%) had thought differently during the reward trials. In the competition trials, 62.07% of the participants thought differently, while 42.86% acted differently. A majority of the participants (68.97%) confirmed that they believed the cover story. When asked to rate the importance of “winning the money” and “the amount of reward” on a scale of 1 to 5, the participants reported an average score of 3.276 for “winning the money” and 2.793 for “the amount of reward” (see Appendix A).

### 3.4. Neuroimaging Results

We found similar activation patterns for hot and cool EF networks, as shown in Figure 3. The regions activated in hot and cool EF by the whole-brain analysis were listed in Table 2. When comparing the competition and no-competition sessions, we found that the activation patterns of both hot and cool EF during competition were statistically identical to those in the no competition condition. No significant clusters were found in either No-competition—Competition or Competition—No-competition contrasts (Figure 3, Table 2).

When a monetary reward was presented (Hot–Cool EF), a cluster involving the right insula, right hippocampus, left caudate nucleus, left ventral lateral nucleus, left superior parietal gyrus, and right superior frontal gyrus–dorsolateral was activated in the no-competition task. On the other hand, in the competition task, Hot–Cool EF activated the right precuneus and caudate nucleus. Further analysis revealed that, in the Hot–Cool EF contrast, greater activation was observed in the occipital gyrus, lingual gyrus, fusiform gyrus, left lobule VI of the cerebellar hemisphere, and left cuneus areas under the no-competition condition compared to the competition condition (Figure 3, Table 2).

### 3.5. Correlation Analysis

When testing the correlation between hot EF and Emotional Control and Self-Monitor, our findings revealed a positive association between a cerebellum cluster under the hot EF condition and the Self-Monitor scale (*p* = 0.016, corrected for multiple comparisons, Figure 4a). When examining Cool EF regions to be related to Working Memory, Shift, or Inhibit, we observed a significant positive correlation between a cluster in the right parietal gyrus and the Working Memory scale (*p* = 0.008, corrected for multiple comparisons). Additionally, a negative trend was found between a cerebellum cluster and the BRIEF-A Shift scale (Figure 4b). For the Hot–Cool EF condition, we identified two trends in cluster 1 (including the right insula, hippocampus, and superior frontal gyrus) which demonstrated a positive correlation with the BART score and a negative correlation with sensitivity to reward (Figure 4c).

## 4. Discussion

This study aimed to provide a systematic framework for the neural basis of hot and cool EF engagement in relation to socio-emotional regulation. We utilized a reward-based n-back task within a social competition game under fMRI to examine the neural correlates of cool and hot executive functioning. The whole-brain analysis of cool EF revealed six significant clusters with the largest cluster located in the left supplementary motor area, insula, middle frontal gyrus, parietal gyrus, pre-/post-central gyrus, and bilateral superior frontal gyrus–dorsolateral. The finding was consistent with previous meta-analyses which identified the cluster of cool EF in the bilateral insula, inferior and middle frontal gyrus, parietal lobule, and the right supplementary motor area [69]. Similarly, our analysis of hot EF showed activations in four significant clusters with the largest cluster (7270 mm^3^) located in the left supplementary motor area, insula, parietal gyrus, bilateral precentral gyrus, and superior frontal gyrus–dorsolateral.

When examining the difference between hot and cool EF, the whole-brain analysis of Hot–Cool EF contrast highlighted an overall functioning of brain areas involved in the reward effect, including the right insula, right hippocampus, superior frontal gyrus–dorsolateral, left caudate nucleus, ventral lateral nucleus, and superior parietal gyrus. Although we did not see an obvious disassociation between the dorsal and ventral networks for cool and hot EF, respectively, the activation in the superior frontal gyrus was consistent with Lee et al. [69] in hot and cool EF, and was found to be associated with higher cognitive functions [70,71,72]. More specifically, regions in the prefrontal cortex are functionally coupled in cognitive control for processing expected rewards and strategy selection while choosing between tasks [11,73,74].

Additionally, our results revealed activation in areas associated with reward processing. A recent study has highlighted that rewards influence the flexible allocation of resources but not capacity in visual working memory [75]. The reward circuit related areas, such as the right insula, hippocampus, caudate nucleus, and ventral lateral nucleus, were also activated in our analysis [76,77]. The insular cortex has been reported to be involved in functional integration [78] and may play a key role in associating visceral sensation and autonomic responses with cognitive appraisal of social or emotional information [79,80]. Nonetheless, our Hot–Cool EF map did not reveal the subcortical regions of the olfactory cortex (extending to the amygdala, caudate, and putamen) as reported in previous studies [69,81]. One possible reason for not detecting more of the emotion (e.g., empathy) related areas could be the amount of reward in our game. From the results of the debriefing form, most of the participants reported that winning money was not very important to them (Appendix A). Therefore, the participants may not have shown a difference in their emotional responses between the hot and cool trials.

Additionally, a cluster of the right insula, hippocampus, caudate nucleus, and superior frontal gyrus was found to correlate with sensitivity to reward. This finding validated the reward effect that we targeted in our fMRI task [61]. The involvement of the hippocampus and prefrontal regions in the reward system has been well-documented in the previous literature [82,83,84]. Basal ganglia activities were modulated by motivation and the caudate nucleus played an important role in affective processing [85,86].

Under the competitive condition, Hot–Cool EF was also found to be associated with greater activation in the right precuneus and caudate nucleus. The precuneus is associated with risk-taking behaviour, whereas the caudate nucleus is linked to sensitivity to reward [87]. The precuneus and caudate nucleus are key nodes of the theory of the mind and the important subcortical motivational regions [88]. Functional connectivity between the caudate nucleus and the superior temporal gyrus and precuneus is highly correlated with social intelligence [89]. Abnormal functional connectivity between the precuneus and hippocampus is also an important marker of cognitive impairment [90]. Thus, higher right precuneus and caudate nucleus activation may lead to enhanced impulsive motivation to pursue rewards, thereby promoting cool EF.

The correlation analysis supported our hypotheses and showed that cool EF activated areas were associated with the BRIEF-A Working Memory and Shift domains. We found that a cluster in the right parietal gyrus was significantly and positively correlated with working memory, while the ability to shift between tasks was related to a cluster in the cerebellum. These findings are consistent with the previous literature on working memory and task switching [91,92,93]. The parietal lobe is an important region for cognitive control and flexible transformation [34,94]. An enhanced switching cost was observed after resection of cerebellar lesions [95]. Moreover, the results showed a positive trend between a cerebellum cluster and the BRIEF-A Self-Monitor domain under hot EF. The cerebellum plays an important role in emotional and social behaviours, such as the ability to recognize the emotions and perspectives of others [96,97,98]. Activation of the right cerebellum has been observed during social recognition in specific contexts [99], indicating lateralization in this brain region. In addition, the cerebellum participates in performance monitoring processes, such as feedback learning and cognitive inhibition [100,101,102]. These results supported the involvement of the cerebellum in strategy adjustment and behavioural monitoring in the processing of hot and cool EF. Noteworthy, no cluster was found to be associated with the BRIEF-A Emotional Control domain. The Emotional Control items in the BRIEF-A primarily assess anger management. However, given the participants’ overall good performance on the task, our design may not have provoked angry emotions, and therefore, may not have activated the relevant brain regions.

Next, Hot–Cool EF activated clusters showed a positive correlation trend with the total score of BART, indicating a link between risk-taking behaviour and the brain regions involved in reward processing. Risk-taking behaviour has been associated with some brain regions, including the insula [103], prefrontal gyrus [56,104], and cerebellum [105], overlapping with the reward-activated regions we detected in our Hot–Cool EF map. Our results were consistent with the previous research suggesting that decision-making is influenced by the balance between the number of rewards and the level of risk [37,106].

Nevertheless, this study has several limitations. We did not find any differences between trials with and without competition in either the hot or cool conditions. This could be explained by analysing the participants’ debriefing responses. We found that, while 68.97% of them believed they were playing with a real competitor, only 42.86% reported using different strategies during competition blocks. The participants generally expressed neutral feelings toward the competitor (Appendix A). Moreover, our current task was designed to link the task reward to real monetary compensation, and to accurately activate a more robust reward related to brain regions [107]. Nonetheless, it is possible that the amount of reward in the current study was not enough to fully motivate all the participants. Future studies may increase the amount of incentive and make the competition more realistic to trigger emotions. Lastly, while we conducted a power analysis to confirm that our sample size provided sufficient statistical power (Appendix A), replicating these findings with a larger sample size would be beneficial.

Interestingly, although no differences were found between competition and no-competition under hot or cool EF, we discovered significant clusters in no-competition versus competition condition in Hot–Cool EF after incorporating the reward condition. These clusters were located in the bilateral occipital gyrus, fusiform gyrus, and lingual gyrus. When examining the pure reward effect of the brain, we found greater activation in the “no competition” trials compared to the competition trials in both the right and left occipital gyrus, fusiform gyrus, and lingual gyrus. These regions were associated with higher visual processing and reading [108,109,110], which would be expected due to the letter stimuli used in our 2-back task. The absence of a competitor in the “no competition” trials may have allowed the participants to focus more on processing stimuli during the task, consistent with a previous study [48].

## 5. Conclusions

In conclusion, we verified the brain regions activated by hot and cool EF and examined their differences. Hot and cool EF activated distinct neural networks (in addition to DLPFC and VMPFC). The results highlighted reward-related neural activation in the right insula, hippocampus, left caudate nucleus, and superior parietal gyrus during the no-competition task, and activated the right precuneus and caudate nucleus in the competition condition. To understand the neural mechanisms underlying executive functions and their relationship to behaviour, we further investigated the correlations between these activated areas and behavioural measures. Using an fMRI task design with real monetary compensation, we identified brain regions associated with reward and executive functions. The current results verified the neural correlates of hot and cool EF with a more robust task design. We also found that the two circuits of hot and cool EF both involved the cerebellum when associated with different domains of executive functions (Self-Monitor and Shift). The distinct networks may work together to regulate our social interactions (as seen in the competition condition) which were associated with cerebellar activation. We suggested that education and learning settings should consider the emotions of learners and environmental influences. The findings provided insights into the underlying mechanism of learning under hot and cool situations and could suggest the consideration of emotions and the influence of the environment to enhance learning. The implications of this research may also help to understand the development and treatment of EF-related disorders.

## Figures and Tables

**Figure 1 sensors-25-00806-f001:**
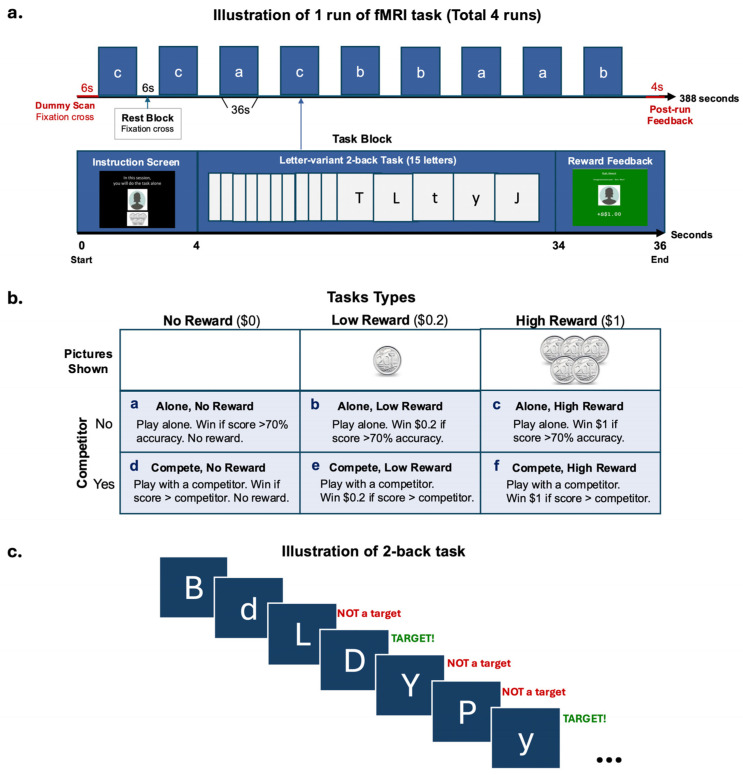
fMRI task design. (**a**) Illustration of one run of the blocked design fMRI task. (**b**) Task types used in each block. (**c**) Illustration of the 2-back task used in a block.

**Figure 2 sensors-25-00806-f002:**
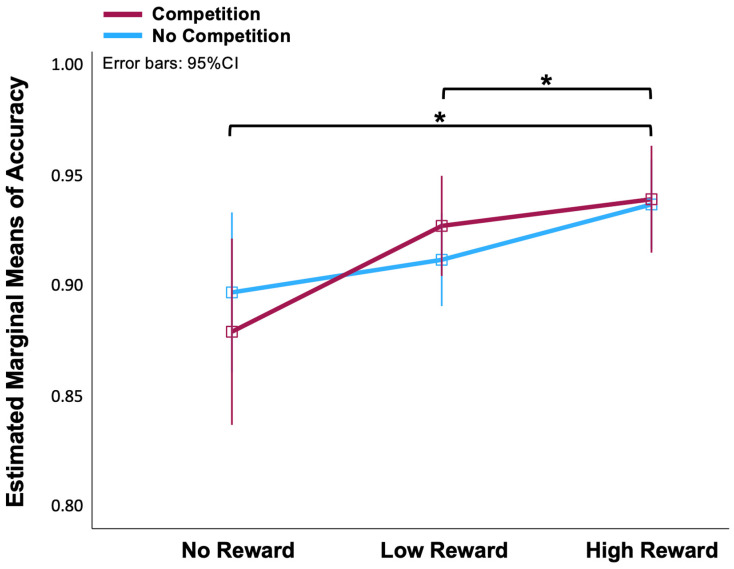
Line chart of means of accuracy in No-, Low-, High-Reward and No-Competition, Competition conditions. * The mean difference is significant at *p* < 0.05 (Bonferroni corrected for multiple comparisons).

**Figure 3 sensors-25-00806-f003:**
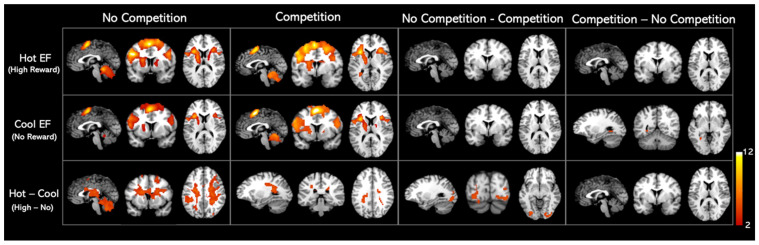
Whole-brain analysis activation maps for Hot and Cool EF in healthy adults. Significant clusters of the whole brain analysis were shown in colour (*p* < 0.001 uncorrected; k > 100 cluster-level corrected; FWE *p* < 0.01 cluster size threshold). The contrasts details are listed in the Appendix A.

**Figure 4 sensors-25-00806-f004:**
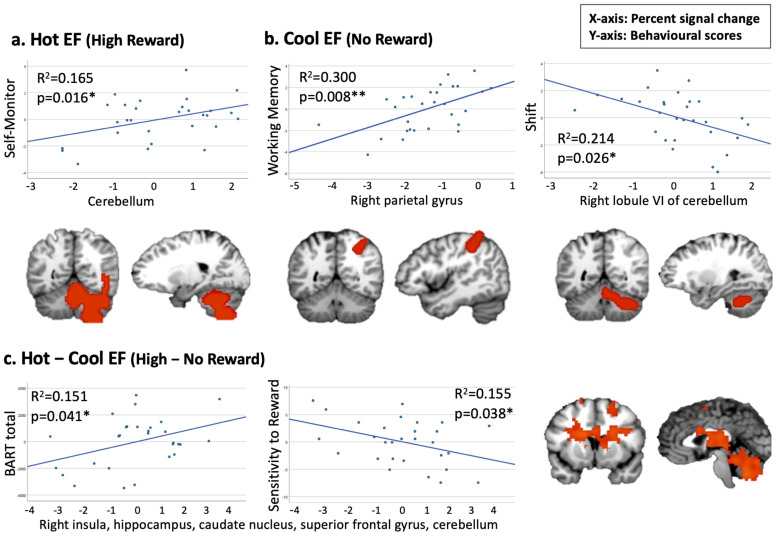
Pearson’s correlations between Hot and Cool EF fMRI clusters and behavioural measures (* *p* < 0.05, ** *p* < 0.01). (**a**) Positive association between Self-Monitor and cluster 3 in the Hot EF condition. (**b**) Positive correlation between Working Memory and cluster 3 in the Cool EF condition; negative association between Shift and cluster 4 in the Cool EF condition. (**c**) Positive association between BART and cluster 1 in the Hot–Cool EF condition; negative association between Sensitivity to Reward and cluster 1 in the Hot–Cool EF condition.

**Table 1 sensors-25-00806-t001:** Demographic and behavioural variables of the participants (*n* = 29, male/female = 14/15).

Measures	Mean (Standard Deviation)
Age (years)	25.55 (4.54)
BART Total (points)	7777.59 (1937.21)
BRIEF-A Inhibit *	51.45 (9.04)
BRIEF-A Shift *	55 (8.73)
BRIEF-A Emotional Control *	50.33 (9.31)
BRIEF-A Self-Monitor *	47.3 (8.65)
BRIEF-A Working Memory *	53.94 (9.21)
BRIEF-A Reward	10.72 (4.17)
Go/No-Go Overall Accuracy	96.7% (2.6%)

The variables are demonstrated as mean (standard deviation). BART, Balloon Analogue Risk Task; BRIEF-A, Behaviour Rating Inventory of Executive Function—adult version (* All scores were converted to T-scores); SPSRQ, The Sensitivity to Punishment and Sensitivity to Reward Questionnaire.

**Table 2 sensors-25-00806-t002:** Significant clusters from the whole brain analysis (uncorrected *p* < 0.001; cluster size k > 100; FWE-corrected *p* < 0.01 (cluster size threshold)).

Condition	Cluster	Volume	Cluster p(FWE)	T	x	y	z	Label
Hot EF (High reward—Rest)								
No Compete	1	7270	4.16 × 10^−11^	15.87	−39	−42	45	Left parietal gyrus excluding supramarginal and angular gyrus
				14.98	−27	−3	57	Left superior frontal gyrus–dorsolateral
				14.28	−30	18	9	Left insula
				13.95	−42	3	30	Left precentral gyrus
				13.82	−6	0	57	Left supplementary motor area
				12.13	−54	−24	45	Left postcentral gyrus
				12.12	−45	−36	42	Left parietal gyrus excluding supramarginal and angular gyrus
				12.10	−27	−57	48	Left superior parietal gyrus
				11.91	30	18	6	Right insula
				11.41	−45	0	42	Left precentral gyrus
				10.46	18	0	63	Right superior frontal gyrus–dorsolateral
				10.27	24	−6	51	Right precentral gyrus
				8.91	45	3	33	Right precentral gyrus
				8.55	9	15	45	Right supplementary motor area
	2	608	0.000001	10.97	−45	−69	−3	Left middle occipital gyrus
				8.38	−42	−51	−30	Left crus I of cerebellar hemisphere
				7.65	−27	−54	−30	Left crus VI of cerebellar hemisphere
				6.48	−42	−84	−6	Left inferior occipital gyrus
				6.14	−33	−90	−9	Left inferior occipital gyrus
	3	2416	2.22 × 10^−16^	6.62	0	−45	−15	Lobule III of vermis
				9.05	42	−54	−33	Right crus I of cerebellar hemisphere
				9.03	30	−48	−33	Right lobule VI of cerebellar hemisphere
				9.02	18	−66	−48	Right lobule VIII of cerebellar hemisphere
				8.74	39	−60	−30	Right crus I of cerebellar hemisphere
				8.71	24	−57	−27	Right lobule VI of cerebellar hemisphere
				8.70	12	−72	−45	Right lobule VII of cerebellar hemisphere
				7.96	15	−48	−21	Right lobule IV–V of cerebellar hemisphere
				7.39	9	−69	−24	Right lobule VI of cerebellar hemisphere
				7.17	6	−66	−27	Lobule VII of vermis
				7.05	51	−69	−6	Right inferior temporal gyrus
				6.58	−24	−66	−51	Left lobule VIII of cerebellar hemisphere
	4	647	7.67 × 10^−7^	9.59	39	−42	48	Right parietal gyrus excluding supramarginal and angular gyrus
				5.38	27	−60	54	Right superior parietal gyrus
				5.16	15	−63	54	Right precuneus
				4.45	30	−66	27	Right superior occipital gyrus
Compete	1	6621	1.72 × 10^−10^	15.01	−39	−42	45	Left parietal gyrus excluding supramarginal and angular gyrus
				12.82	−6	0	60	Left supplementary motor area
				12.79	−27	−57	48	Left superior parietal gyrus
				12.5	−27	−3	57	Left superior frontal gyrus–dorsolateral
				12.46	−45	0	30	Left precentral gyrus
				12.39	−30	15	9	Left insula
				12.32	−3	3	57	Left supplementary motor area
				12.06	−24	−6	60	Left superior frontal gyrus–dorsolateral
				11.66	−48	−3	45	Left precentral gyrus
				11.43	−6	9	51	Left supplementary motor area
				9.9	24	−3	54	Right precentral gyrus
				9.26	15	3	57	Right supplementary motor area
				7.92	−39	−54	−30	Left crus I of cerebellar hemisphere
	2	1719	8.22 × 10^−14^	9.19	18	−51	−24	Right lobule IV–V of cerebellar hemisphere
				9.16	30	−48	−30	Right lobule VI of cerebellar hemisphere
				9.08	3	−57	−12	Lobule IV–V of vermis
				8.75	27	−57	−27	Right lobule VI of cerebellar hemisphere
				8.43	12	−72	−45	Right lobule VIII of cerebellar hemisphere
				8.1	0	−45	−21	Lobule III of vermis
				8.02	18	−63	−48	Right lobule VIII of cerebellar hemisphere
				7.74	0	−45	−15	Lobule III of vermis
				7.42	3	−66	−33	Lobule III of vermis
				6.37	−21	−69	−51	Right lobule VIII of cerebellar hemisphere
	3	591	8.56 × 10^−7^	8.93	36	−45	48	Right parietal gyrus excluding supramarginal and angular gyrus
				7.75	45	−36	48	Right parietal gyrus excluding supramarginal and angular gyrus
	4	187	0.004	8.05	45	−66	−6	Right inferior temporal gyrus
				6.10	45	−81	−6	Right inferior occipital gyrus
Cool EF (No reward—Rest)								
No compete	1	4056	3.38 × 10^−10^	14.61	−6	0	57	Left supplementary motor area
				11.35	−27	−3	57	Left superior frontal gyrus–dorsolateral
				9.99	−30	18	9	Left insula
				9.85	−45	−33	45	Left parietal gyrus excluding supramarginal and angular gyrus
				9.57	−54	−21	48	Left postcentral gyrus
				9.48	−42	0	30	Left precentral gyrus
				9.31	−48	0	42	Left precentral gyrus
				9.06	−39	−42	45	Left parietal gyrus excluding supramarginal and angular gyrus
				7.87	−27	−54	45	Left parietal gyrus excluding supramarginal and angular gyrus
				7.03	−54	−18	24	Left postcentral gyrus
				7.01	−48	−30	60	Left postcentral gyrus
				6.81	18	0	63	Right superior frontal gyrus–dorsolateral
				5.76	−39	27	33	Left middle frontal gyrus
	2	473	0.00003	7.07	30	−48	−30	Right lobule VI of cerebellar hemisphere
				7.00	42	−54	−33	Right crus I of cerebellar hemisphere
				6.76	39	−60	−30	Right crus I of cerebellar hemisphere
				5.87	15	−51	−21	Right lobule IV–V of cerebellar hemisphere
				4.90	6	−54	−15	Lobule IV–V of vermis
				4.58	0	−45	−15	Lobule III of vermis
	3	297	0.001	7.02	21	−66	−51	Right lobule VIII of cerebellar hemisphere
				6.97	24	−69	−54	Right lobule VIII of cerebellar hemisphere
				4.97	6	−78	−45	Right lobule VIIB of cerebellar hemisphere
	4	176	0.009	7.00	−42	−69	−3	Left middle occipital gyrus
				5.64	−42	−84	−6	Left inferior occipital gyrus
	5	376	0.0002	6.80	30	18	9	Right insula
				6.12	48	3	30	Right precentral gyrus
				6.02	51	6	21	Right precentral gyrus
	6	280	0.001	6.40	48	−33	48	Right parietal gyrus excluding supramarginal and angular gyrus
				5.92	39	−42	48	Right parietal gyrus excluding supramarginal and angular gyrus
				4.78	45	−45	60	Right superior parietal gyrus
				4.61	30	−54	45	Right parietal gyrus excluding supramarginal and angular gyrus
Compete	1	5661	3.39 × 10^−10^	14.6	−3	3	57	Left supplementary motor area
				14.38	−3	9	54	Left supplementary motor area
				14.28	−6	0	60	Left supplementary motor area
				13.83	−9	−3	63	Left supplementary motor area
				12.67	−27	−3	54	Left superior frontal gyrus–dorsolateral
				11.32	−33	15	9	Left insula
				10.71	−45	−3	42	Left precentral gyrus
				10.51	−48	−33	48	Left postcentral gyrus
				10.34	−45	0	30	Left precentral gyrus
				9.9	27	0	51	Right precentral gyrus
				9.78	−54	6	30	Left precentral gyrus
				9.64	−36	−42	48	Left parietal gyrus excluding supramarginal and angular gyrus
				9.32	−51	3	21	Left precentral gyrus
				9.06	−27	−57	48	Left superior parietal gyrus
	2	1229	1.3 × 10^−11^	9.20	30	−51	−30	Right lobule VI of cerebellar hemisphere
				8.13	3	−57	−12	Lobule IV–V of vermis
				8.01	39	−60	−30	Right crus I of cerebellar hemisphere
				7.76	18	−60	−45	Right lobule VIII of cerebellar hemisphere
				7.64	21	−66	−48	Right lobule VIII of cerebellar hemisphere
				6.11	12	−45	−24	Right lobule III of cerebellar hemisphere
				4.11	3	−75	−30	Lobule VII of vermis
	3	190	0.002	9.17	−39	−54	−30	Left crus I of cerebellar hemisphere
	4	450	0.000006	7.66	51	−33	48	Right parietal gyrus excluding supramarginal and angular gyrus
				7.49	48	−36	51	Right parietal gyrus excluding supramarginal and angular gyrus
				7.37	36	−42	45	Right parietal gyrus excluding supramarginal and angular gyrus
				5.89	30	−54	45	Right parietal gyrus excluding supramarginal and angular gyrus
				4.06	15	−63	54	Right precuneus
	5	228	0.001	7.62	−42	−69	0	Left middle occipital gyrus
				4.61	−42	−84	−6	Left inferior occipital gyrus
	6	170	0.004	6.97	45	−66	−6	Right inferior temporal gyrus
				5.71	42	−84	−6	Right inferior occipital gyrus
				5.54	48	−78	−3	Right inferior occipital gyrus
Hot—Cool EF								
No Compete	1	12,249	0.0003	7.87	5.67	33	3	Right insula
				7.06	30	−36	3	Right hippocampus
				6.66	−15	15	12	Left caudate nucleus
				6.6	−12	−15	18	Left ventral lateral nucleus
				6.48	−30	−51	60	Left superior parietal gyrus
				6.42	24	−9	60	Right superior frontal gyrus–dorsolateral
				6.19	−21	−57	63	Left superior parietal gyrus
Compete	1	392	0.001	4.78	18	−54	45	Right precuneus
				3.59	15	12	21	Right caudate nucleus
No Compete—Compete	1	387	0.0004	5.27	33	−87	−6	Right inferior occipital gyrus
				5.23	27	−90	9	Right middle occipital gyrus
				4.53	21	−81	−6	Right lingual gyrus
				4.35	36	−57	−18	Right fusiform gyrus
				3.97	30	−69	−18	Right fusiform gyrus
	2	216	0.007	4.86	−30	−78	−6	Left fusiform gyrus
				4.41	−24	−87	15	Left middle occipital gyrus
				4.25	−21	−90	12	Left superior occipital gyrus
				3.84	−21	−78	−18	Left lobule VI of cerebellar hemisphere
				3.77	−12	−90	15	Left cuneus

## Data Availability

The data presented in this study are available at https://doi.org/10.21979/N9/U07BP3.

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
