# Peer review of "A Functional Magnetic Resonance Imaging Investigation of Hot and Cool Executive Functions in Reward and Competition"

_sensors, 2025, doi:10.3390/s25030806_

Round 1

Reviewer 1 Report

Comments and Suggestions for Authors

In this meta analysis (sensors-3316328), the authors used functional magnetic resonance imaging (fMRI) to determine the role of hot and cool executive functioning (EF) in reward processing and its relationship to performance under social competition. The results are complete based on the literature review, and have reference value, thus it can be accepted after addressing some technical issues.

1.      Carefully check the format of the journal.

2.      Title: “An” is incorrect. Additionally, it is recommended to use full spelling instead of “fMRI”.

3.      Keywords: The first letter of each keyword does not need to be capitalized.

4.      The title and author information need to be provided in the supporting material files.

5.      Introduction: It is recommended to discuss more detection techniques to align with the special issue and scope of this special issue.

6.      The citation format does not meet the requirements. “[Number]”.

7.      Subtitles: Except for prepositions, articles, and conjunctions, the first letters of all other words need to be capitalized.

8.      The reference format does not meet the requirements. In addition, some reference information is incomplete, such as year, volume number, and page number.

9.      Check and improve English writing.

Comments on the Quality of English Language

Check and improve English writing.

Author Response

Thank you very much for taking the time to review this manuscript. Please find the detailed responses below (a PDF version is attached) and the corresponding revisions in the re-submitted files. 

  1. Carefully check the format of the journal.

Thank you for the comment. We have carefully followed the format of the journal and applied it to the manuscript. We have downloaded the MDPI Endnote style file from the SENSOR website and applied it to the manuscript.

  1. Title: “An” is incorrect. Additionally, it is recommended to use full spelling instead of “fMRI”.

Thank you for the suggestion. We have edited the title to: A Functional Magnetic Resonance Imaging Investigation of Hot and Cool Executive Functions in Reward and Competition

  1. Keywords: The first letter of each keyword does not need to be capitalized.

Thank you for the suggestion. We have corrected the keywords.

  1. The title and author information need to be provided in the supporting material files.

Thank you for the suggestion. We have added the title and author information in the supporting material file.

  1. Introduction: It is recommended to discuss more detection techniques to align with the special issue and scope of this special issue.

Thank you for your comment. We have provided additional details about the fMRI technique and image processing steps in the Introduction and Materials and Methods sections

Introduction section, line 103:

“The present study integrated real monetary rewards within a task-based fMRI design to investigate these associations, providing a more direct and tangible incentive for participants [26,38,60,61]. We examined functional activation patterns of social reward processing in healthy adults using task-based fMRI, a non-invasive neuroimaging technique that measures brain activity by detecting changes in blood oxygenation levels, known as the Blood Oxygenation Level Dependent (BOLD) signal. This technique allows for the detection of neural activity in response to cognitive tasks, facilitating the investigation of executive functions and reward processing. We also explored the neural correlates of behavioural measures of hot and cool EF and reward sensitivity.”

Materials and Methods section > Data analysis, line 122:

“MRI data preprocessing and whole-brain univariate analysis were conducted using SPM 12 (Statistical Parametric Mapping software, SPM; Wellcome Department of Imaging Neuroscience, London, U.K. www.fil.ion.ucl.ac.uk/spm) under MATLAB R2020a (The Mathworks, Inc., Natick, MA, USA). The quality of images was carefully checked by the researchers to ensure there were no excessive motion artifacts (translational movement < 3 mm in any direction and rotational movement < 2°). All functional images were corrected for slice timing with the middle slice in the acquisition order as the reference, realigned to the first image to correct for head movement. The individual's T1‐weighted image was coregistered to the mean functional image and then segmented. Functional and anatomical images were normalized to the MNI (Montreal Neurological Institute) space using the DARTEL procedure [54]. A group template was first generated using the participants’ gray matter and white matter masks for more accurate inter-subject alignment. All the functional and anatomical images were registered to the group template and normalized to the MNI space. Functional images were spatially smoothed with a Gaussian kernel of 8 × 8 × 8 mm.”

  1. The citation format does not meet the requirements. “[Number]”.

Thank you for the comment. We have downloaded the MDPI Endnote style file from the SENSOR website and applied it to the manuscript.

  1. Subtitles: Except for prepositions, articles, and conjunctions, the first letters of all other words need to be capitalized.

Thank you for your comment. We have corrected them.

  1. The reference format does not meet the requirements. In addition, some reference information is incomplete, such as year, volume number, and page number.

Thank you for the comment. We have downloaded the Endnote files and Endnote style file from the SENSOR website and applied it to the manuscript.

  1. Check and improve English writing.

Thank you for the comment. We have professionally checked the English writing.

Reviewer 2 Report

Comments and Suggestions for Authors

The authors used functional magnetic resonance imaging to determine the role of hot and cool executive functioning in reward processing and its relationship to performance under social competition. 

The paper has the following strong points:

1) It is well organized and easy to follow.

2) The study used real data coming from participants in several universities.

3) The study methodology is well described through illustrations and examples.

4) The idea behind the study is totally novel and few studies shown similar works.

5) The obtained results seem interesting and promising.

However, the paper has the following comments that should be addressed:

6) Abstract and conclusion should be supported by results and main findings of the study.

7) Please provide more information about the inclusion and exclusion criteria when selecting the participants.

8) I miss more details about the data preprocessing step, e.g. missing/null values, normalization is applied or not, correlation between features is existing or not, etc.

9) Some used references are too old (more than 7 years). It is advised toad more recent references.

Author Response

Thank you very much for taking the time to review this manuscript. Please find the detailed responses below (a PDF version is attached) and the corresponding revisions in the re-submitted files. 

The authors used functional magnetic resonance imaging to determine the role of hot and cool executive functioning in reward processing and its relationship to performance under social competition.

The paper has the following strong points:

1) It is well organized and easy to follow.

2) The study used real data coming from participants in several universities.

3) The study methodology is well described through illustrations and examples.

4) The idea behind the study is totally novel and few studies shown similar works.

5) The obtained results seem interesting and promising.

Thank you for the encouraging comments.

However, the paper has the following comments that should be addressed:

6) Abstract and conclusion should be supported by results and main findings of the study.

Thank you for pointing this out. We have revised the abstract and conclusion accordingly.

We have included the following in the abstract:

“Hot and cool EF activated distinct networks in the right insula, hippocampus, left caudate nucleus, and superior parietal gyrus during the no-competition task, while they differentially activated the right precuneus and caudate nucleus in the competition condition. Further analysis revealed correlations between the Hot-Cool network and reward sensitivity and risk-taking behaviour.”

We have included the following in the conclusion:

“In conclusion, we have verified the brain regions activated by hot and cool EF and examined their differences. Hot and cool EF activated distinct neural networks (in addition to DLPFC and VMPFC). The results highlighted reward-related neural activation in the right insula, hippocampus, left caudate nucleus, and superior parietal gyrus during the no-competition task, and activated the right precuneus and caudate nucleus in the competition condition. To understand the neural mechanisms underlying executive function and their relationship to behaviour, we further investigated the correlations between these activated areas and behavioural measures. Using an fMRI task design with real monetary compensation, we identified brain regions associated with reward and executive function. The current results verified the neural correlates of hot and cool EF with a more robust task design. We also found that the two circuits of hot and cool EF both involved the cerebellum when associated with different domains of executive functions (Self-Monitor and Shift).”

7) Please provide more information about the inclusion and exclusion criteria when selecting the participants.

Thank you for the comment. We have revised the Materials and Methods and included more information regarding the criteria under the Participants section, line 124:

“A total of 33 right-handed healthy adults between ages of 21 to 40 were initially recruited through poster advertisements at Nanyang Technological University and National University of Singapore. The study was approved by the respective institutional review boards (NHG DSRB Ref 2017/01125; IRB-2016-01-003) in these two universities. All participants provided written informed consent before participation and were screened to rule out (1) neurological or psychiatric disorders, (2) irremovable metallic objects or implants (e.g., pacemaker), and (3) other factors that increase the risk of having an adverse event during an MRI scan. Four subjects were excluded from the analyses due to scan incompletion, incidental findings (e.g., arachnoid cysts) or excessive motion artifacts, and 29 healthy participants (Male/Female: 14/15, Age Mean/Std=25.55/4.54) were included in the final analyses. The demographic information for the subjects is listed in Table 1.”

8) I miss more details about the data preprocessing step, e.g. missing/null values, normalization is applied or not, correlation between features is existing or not, etc.

Thank you for the comment. We have revised the Materials and Methods under the Data Analysis section, line 246 and included the following information:

“MRI data preprocessing and whole-brain univariate analysis were conducted using SPM 12 (Statistical Parametric Mapping software, SPM; Wellcome Department of Imaging Neuroscience, London, U.K. www.fil.ion.ucl.ac.uk/spm) under MATLAB R2020a (The Mathworks, Inc., Natick, MA, USA). The qualities of structural and functional images were checked carefully by the researchers. All functional images were corrected for slice timing with the middle slice in the acquisition order as the reference, realigned to the first image to correct for head movement. The individual's T1‐weighted image was coregistered to the mean functional image and then segmented. Functional and anatomical images were normalized to the MNI (Montreal Neurological Institute) space using the DARTEL procedure [54]. A group template was first generated using the participants’ gray matter and white matter masks for more accurate inter-subject alignment. All the functional and anatomical images were registered to the group template and normalized to the MNI space. Functional images were spatially smoothed with a Gaussian kernel of 8 × 8 × 8 mm.”

9) Some used references are too old (more than 7 years). It is advised to add more recent references.

Thank you for pointing this out. We have carefully reviewed the references and updated them to include more recent studies. As a result, 52% of the references now fall within the past seven years.

Reviewer 3 Report

Comments and Suggestions for Authors

·       The novelty of the research should be highlighted clearly.

·       How proposed method is unique compared to the previous studies should be clarified.

·       The problem is relevant but should be explained more deeply with references to prior research gaps.

·       Provide a rationale for the reward values used in the task and describe how motion artifacts were dealt within MRI data.

·        Justify how 29 samples can give a reliability to the proposed method.

·       Explain the importance of the significant brain clusters and compare findings with similar studies.

·       include more clarity in the visuals-graphs or figures-to summarize main results.

·       Highlight the limitations of the proposed method.

Author Response

Thank you very much for taking the time to review this manuscript. Please find the detailed responses below (a PDF version with figures is attached) along with the corresponding revisions in the re-submitted files.

 The novelty of the research should be highlighted clearly.

  • How proposed method is unique compared to the previous studies should be clarified.

Thank you for the comment. We have revised the Introduction section, line 103:

“The present study integrates real monetary rewards within a task-based fMRI design to investigate these associations, providing a more direct and tangible incentive for participants [26,38,60,61]. We examine functional activation patterns of social reward processing in healthy adults using task-based fMRI, a non-invasive neuroimaging technique that measures brain activity by detecting changes in blood oxygenation levels, known as the Blood Oxygenation Level Dependent (BOLD) signal. This technique allows for the detection of neural activity in response to cognitive tasks, facilitating the investigation of executive functions (EF) and reward processing. We also explored the neural correlates of behavioural measures of hot and cool EF and reward sensitivity.”

  • The problem is relevant but should be explained more deeply with references to prior research gaps.

Thank you for the comment. We have revised these two paragraphs in the Introduction section, line 52 and line 100, and incorporated more recent references in the Introduction section to make the details more explicit.

“Previous studies engaging executive function, such as response inhibition [20-23], cognitive flexibility [10,24], attention [25], or working memory [26,27] have shown performance to differ under different reward conditions (i.e. with vs. without reward incentives). However, they mostly examined cool and hot EF in isolated conditions. This limits our understanding of how the cortical activity related to hot and cool EF interacts to modulate behavioral responses during the “hot” context of emotional involvement and reward feedback.”

And

“Although there are fMRI studies examining hot and cool EF [27,59], no study has yet determined how the hot and cool networks are associated with the influence of reward on executive control and its relationship with cognitive performance, particularly in the context of social competition. The present study integrates real monetary rewards within a task-based fMRI design to investigate these associations, providing a more direct and tangible incentive for participants [26,38,60,61]. We examine functional activation patterns of social reward processing in healthy adults using task-based fMRI, a non-invasive neuroimaging technique that measures brain activity by detecting changes in blood oxygenation levels, known as the Blood Oxygenation Level Dependent (BOLD) signal. This technique allows for the detection of neural activity in response to cognitive tasks, facilitating the investigation of executive functions (EF) and reward processing. We also explored the neural correlates of behavioural measures of hot and cool EF and reward sensitivity.”

  • Provide a rationale for the reward values used in the task and describe how motion artifacts were dealt within MRI data.

Thank you for the valuable comment. Our current task was designed to connect the task reward to real monetary compensation received by the participant rather than abstract game points. We believed that the design is more robust in activating the related reward regions in the brain. We maximized the perception of the range or reward types, with participants seeing a blank picture for no reward (SGD0.00), a 20-cent coin for low reward (SGD0.20) or five 20-cent coins for high reward (SGD1.00), respectively. Due to our budget constraint, we could only provide a maximum reward of SGD14.40. Thus, we acknowledge that the overall reward amount used in this study may not have been sufficient to fully motivate all participants and have raised this as a limitation of the current study.

We have included the above information in Figure 1b and the Materials and Methods,  line 200, as extracted below:

“The amount of money associated with the reward types were SGD0.00, SGD0.20, and SGD1.00, with participants seeing a blank picture for no reward, a 20-cent coin for low reward, or five 20-cent coins for high reward, respectively. Participants had the opportunity to win and receive a maximum amount of SGD 14.40 as real monetary compensation.”

We have also revised this limitation in the Discussion section, line 448, to make it clearer as below.

“Moreover, our current task was designed to link the task reward to real monetary compensation and to accurately activate a more robust reward related brain regions [107]. Nonetheless, it is possible that the amount of reward in the current study was not enough to fully motivate all the participants. Future studies may increase the amount of incentive and make the competition more realistic to trigger emotions.”

Regarding motion artifacts, we have included a paragraph in the Materials and Methods under the Data Analysis section, line 250 on how motion artefacts were addressed.

“The quality of images was carefully checked by the researchers to ensure there were no excessive motion artifacts (translational movement < 3 mm in any direction and rotational movement < 2°). All functional images were corrected for slice timing with the middle slice in the acquisition order as the reference and realigned to the first image to correct for head movement.”

  • Justify how 29 samples can give a reliability to the proposed method.

Thank you for the question. A power analysis was conducted using fmripower (http://fmripower.org/) to estimate the required sample size for detecting significant effects in our fMRI design. The analysis, with a Type I error rate of 0.05, indicated that including more than 28 subjects would achieve a desired power range of 70% to 99% for reliably detecting group-level effects (Supplementary Figure S2). Furthermore, the task design (blocked design with 9 blocks, 3 reward types, and 4 runs) and replication of each condition across two runs ensure robust and reliable measurements. Previous studies with similar task designs and sample sizes have successfully detected reliable brain activation patterns in reward-related tasks [1-3], further supporting the adequacy of our sample size. However, it would be beneficial for these findings to be replicated with a larger sample size, which we have acknowledged as a limitation in the Discussion section.

This has been included in the following paragraph in the discussion section, line 453, and the Figure S2 is now included in the supplementary data.

“Lastly, while we conducted a power analysis to confirm that our sample size provided sufficient statistical power (Supplementary Figure S2), replicating these findings with a larger sample size would be beneficial.”

  • Explain the importance of the significant brain clusters and compare findings with similar studies.

Thank you for your comment. We have included more information and updated the references in the Discussion section.

  • include more clarity in the visuals-graphs or figures-to summarize main results.

Thank you for your comment. We have included more information and made the figures clearer (Figure 1, 3 and 4).

  • Highlight the limitations of the proposed method.

Thank you for your comment. We have included a discussion of the limitations related to the task design and sample size in the Discussion section, line 443.

“Nevertheless, this study had several limitations. We did not find any differences between trials with and without competition in either hot or cool conditions. This could be explained by the analysis of the participants' debriefing responses. Where we found that while 68.97% of them believed they were playing with a real competitor, only 42.86% reported using different strategies during competition blocks. The participants generally expressed neutral feelings for the competitor (Supplementary Table S2). Moreover, our current task was designed to link the task reward to real monetary compensation and to activate more robust reward related brain regions [91,92]. Nonetheless, it is possible that the amount of reward in the current study was not enough to fully motivate all the participants. Future studies may want to increase the amount of incentive and make the competition more emotionally engaging. Lastly, while we conducted a power analysis to confirm that our sample size provides sufficient statistical power (Supplementary Figure S2), replicating these findings with a larger sample size would be beneficial.”

References:

  1. Beck, S.M.; Locke, H.S.; Savine, A.C.; Jimura, K.; Braver, T.S. Primary and secondary rewards differentially modulate neural activity dynamics during working memory. PLoS One 2010, 5, e9251, doi:10.1371/journal.pone.0009251.
  2. Pochon, J.B.; Levy, R.; Fossati, P.; Lehericy, S.; Poline, J.B.; Pillon, B.; Le Bihan, D.; Dubois, B. The neural system that bridges reward and cognition in humans: an fMRI study. Proc Natl Acad Sci U S A 2002, 99, 5669-5674, doi:10.1073/pnas.082111099.
  3. Rademacher, L.; Krach, S.; Kohls, G.; Irmak, A.; Gründer, G.; Spreckelmeyer, K.N. Dissociation of neural networks for anticipation and consumption of monetary and social rewards. Neuroimage 2010, 49, 3276-3285, doi:10.1016/j.neuroimage.2009.10.089.

Round 2

Reviewer 2 Report

Comments and Suggestions for Authors

The authors addressed all my comments and concerns. The quality of the paper is highly improved. I suggest the publication of the paper in the journal.

Reviewer 3 Report

Comments and Suggestions for Authors

none